# Predictors of Voriconazole Trough Concentrations in Patients with Child–Pugh Class C Cirrhosis: A Prospective Study

**DOI:** 10.3390/antibiotics10091130

**Published:** 2021-09-20

**Authors:** Yichang Zhao, Jingjing Hou, Yiwen Xiao, Feng Wang, Bikui Zhang, Min Zhang, Yongfang Jiang, Jiakai Li, Guozhong Gong, Daxiong Xiang, Miao Yan

**Affiliations:** 1The Second Xiangya Hospital, Central South University, Changsha 410011, China; zhaoyichang@csu.edu.cn (Y.Z.); houjingjing2020@csu.edu.cn (J.H.); xiaoyw2005@163.com (Y.X.); roitech@126.com (F.W.); bikui_zh@126.com (B.Z.); zhangmin001@csu.edu.cn (M.Z.); 503060@csu.edu.cn (Y.J.); Lijkai930@163.com (J.L.); gongguozhong@csu.edu.cn (G.G.); xiangdaxiong@csu.edu.cn (D.X.); 2Institute of Clinical Pharmacy, Central South University, Changsha 410011, China; 3Department of Infection, Central South University, Changsha 410011, China

**Keywords:** Child–Pugh C cirrhosis, voriconazole, trough concentrations, administration, *CYP2C19*

## Abstract

This prospective observational study aimed to clinically describe voriconazole administrations and trough concentrations in patients with Child–Pugh class C and to investigate the variability of trough concentration. A total of 144 voriconazole trough concentrations from 43 Child–Pugh class C patients were analyzed. The majority of patients (62.8%) received adjustments. The repeated measured trough concentration was higher than the first and final ones generally (median, 4.33 vs. 2.99, 3.90 mg/L). Eight patients with ideal initial concentrations later got supratherapeutic with no adjusted daily dose, implying accumulation. There was a significant difference in concentrations among the six groups by daily dose (*p* = 0.006). The bivariate correlation analysis showed that sex, CYP2C19 genotyping, daily dose, prothrombin time activity, international normalized ratio, platelet, and Model for end-stage liver disease score were significant factors for concentration. Subsequently, the first four factors mentioned above entered into a stepwise multiple linear regression model (variance inflation factor <5), implying that *CYP2C19* testing makes sense for precision medicine of Child–Pugh class C cirrhosis patients. The equation fits well and explains the 34.8% variety of concentrations (R^2^ = 0.348). In conclusion, it needs more cautious administration clinically due to no recommendation for Child–Pugh class C patients in the medication label. The adjustment of the administration regimen should be mainly based on the results of repeated therapeutic drug monitoring.

## 1. Introduction

Patients with Child–Pugh class C (CP-C) cirrhosis, the most serious class of liver cirrhosis in the Child–Pugh classification [1,2,3], are at a high risk of invasive fungal infections (IFIs) [4,5]. IFIs in CP-C patients are probably caused by their immunocompromised state and the application of a large number of steroids and broad-spectrum antimicrobial agents, as well as invasive procedures [6]. It takes a heavy toll on outcomes with high in-hospital mortality rates if not promptly recognized and treated [7,8], ascribed to the rapid progression with a lung infection and even septic shock [6,9,10].

Voriconazole is a broad-spectrum triazole antifungal agent with good in-body activity against fungal pathogens [11]. It is commonly used clinically to prevent and treat IFIs and is recommended as a first-line drug for invasive aspergillosis by the Infectious Diseases Society of America (IDSA) and the Clinical Pharmacogenetics Implementation Consortium (CPIC) [11,12]. The recommended targeted concentrations are between 1.0–5.5 mg/L according to British guidelines and other studies [13,14].

Voriconazole is metabolized primarily by the hepatic cytochrome P450 (CYP) 2C19; also a substrate for CYP3A4 and CYP2C9 enzyme in the liver (98%) [15], and then excreted through the kidney and bile, with less than 2% of the dose of voriconazole excreted unchanged in the urine through renal excretion [16]. The gene polymorphism of *CYP2C19* and its mediated metabolism can be saturated within the therapeutic concentration range, resulting in the greater individual differences of pharmacodynamic parameters [17]. Voriconazole displays non-linear pharmacokinetic (PK) in adults and manifests extreme inter- and intra-individual PK variability in all patient populations, which are also associated with liver dysfunction, drug–drug interaction, etc. [18,19,20,21]. Although under the standard administration scheme, the voriconazole trough concentration [18], related to clinical efficacy, is distributed in the range of less than 0.2 mg/L to 12 mg/L, even exceptionally high (17.5 mg/L), leading to adverse reactions [22,23]. The medication label of voriconazole recommends that individuals with mild to moderate cirrhosis (Child–Pugh Class A and B) receive the same loading dose as individuals with hepatic function, but half the maintenance dose, while no recommendation is given for individuals with CP-C cirrhosis. The voriconazole terminal half-life (t_1/2_) of CP-C patients in ICU was reported to be increased about five times, resulting in high plasma exposure [24]. All in all, it is particularly urgent and vital to describe the use and trough concentration of voriconazole in CP-C patients.

This prospective observational study aimed to describe voriconazole administrations and trough concentrations in patients with Child–Pugh class C clinically and investigate the variability of trough concentration.

## 2. Results

### 2.1. Patient Characteristics

Forty-three CP-C Asian patients were eligible to participate in the study (Figure 1a). Clinical characteristics are summarized in Table 1. The female-to-male ratio was 4/39. The mean patient age and weight were 49.35 ± 11.65 years (range, 32 to 89 years) and 61.27 ± 12.87 kg (range, 36 to 99 kg). The median duration of treatment with voriconazole was 12 days (range, 5 to 45 days). The median sampling time of C_0_ was 5 days (interquartile range (IQR), 3 to 5 days; range, 3 to 11 days). 15, 23, 5 out of 43 patients (34.9%, 53.5%, 11.6%) took voriconazole orally, intravenously or in sequential therapy before measuring C_0_, respectively. Moreover, a total of 23.3% (10/43) of patients received voriconazole transformationally from intravenously to orally. During the voriconazole therapy period, a total of 37 (86.0%) patients concomitantly used CYP2C19 inhibitors (31/43, 72.1%), antimicrobial agents (25/43, 58.1%), or CYP3A4 inhibitors (1/43, 2.3%), including proton pump inhibitors, antibacterial and antifungal agents. Adverse events identified as voriconazole related were reported in 20 patients (46.5%), including dizziness, hallucinations and visual disturbance such as altered color discrimination, blurred vision and photophobia. The median duration from voriconazole initiation to onset of adverse events was 2 days (range, 1 to 12 days).

### 2.2. CYP2C19 Genotyping

*CYP2C19*1/*1* was the most commonly identified genotype (20, 46.5%), followed by *CYP2C19*1/*2* (13, 30.2%), *CYP2C19*2/*2* (6, 14.0%), *CYP2C19*1/*3* (3, 7.0%) and *CYP2C19*1/*17* (1, 2.3%) (Table 1). For subsequent analysis, *CYP2C19 *1/*17* was analysis appropriately together with *CYP2C19*1/*1* so that *CYP2C19* phenotypes were classified into four categories, **1/*1*, **1/*2*, **1/*3* and **2/*2*.

### 2.3. Administration Dosage of Voriconazole

Table 2 shows that the loading dosage was given in 51.2% patients and 400 mg/12 h, 200 mg/12 h or 200 mg/24 h were the most common administrations (5/22 [22.7%], 4/22 [18.2%], 4/22 [18.2%], respectively). On the other hand, the maintenance dosage of 100 mg/24 h was the major regimen (10, 23.3%). As for the adjustment, the majority of patients (27/43 [62.8%]) received adjusted administration, especially 25.6% once (11/43), followed by three or more times (9, 21.9%) and two times (7, 16.3%).

### 2.4. Voriconazole Trough Concentrations

During the therapy period of voriconazole, a total of 144 plasma concentrations from 43 patients were analyzed in the present study, as shown in Table 1 and Figure 1b, and shown by scatterplot and box plot in Figure 2. The first trough concentration after voriconazole therapy (C_0_), the repeated measured one (C_1_) and the final one (C_ss_) were 2.99 mg/L (IQR, 1.61–5.00 mg/L; range, 0.32–14.08 mg/L, *n* = 43), 4.33 mg/L (IQR, 3.0775–6.1000 mg/L; range,1.86–11.83 mg/L, *n* = 64), 3.90 mg/L (IQR, 2.51–4.84 mg/L; range, 0.60–10.70 mg/L, *n* = 37), respectively. The median sampling time of C_0_ was 5 days (IQR, 3 to 5 days; range, 3 to 11 days) after voriconazole initial therapy, respectively.

A total of 12 patients did not achieve the ideal target trough concentration with 75% of C_0_ over the upper target concentration, whose adjusting progress (range, 2–10 days; mode, two days) with 50 C_min_s in total, shown in Figure 3. On the other hand, for 31 (72.1%) patients with the targeted C_0_, C_min_s showed an upward trend. Among them, 8 patients (8/31, 25.8%) were supratherapeutic levels for concentration, whereas no adjusted daily dose in the subsequent therapy (Figure 4). Besides, only 4 patients (4/43, 9.3%) always got the targeted C_min_s, while a total of 27 patients (62.8%) appeared unideal ones once or more and then accepted adjustment, in which 23 patients were cut the dosage, and the remaining 4 patients experienced dose fluctuations. Finally, there were 66.6% (8/12), 62.5% (5/8), and 83.8% (31/37) patients with C_ss_ achieving the therapeutic target after adjustment, respectively.

The result of the Student’s *t*-test showed that there were no significant differences in the corresponding concentration between 50 mg and 100 mg (*p* = 0.129), 150 mg and 200 mg (*p* = 0.400), so we incorporated them into the same group. Further results showed the significant difference among the three groups by daily dose approximately (*p* = 0.003) (Figure 4). Compared to those of 200 mg, the concentrations were significantly lower of 100 mg (*p* = 0.041) and higher of 400 mg or more (*p* = 0.003) (Figure 5).

### 2.5. Factors Affecting Voriconazole Trough Concentration

It might be adjusted several times or withdrawn when the voriconazole trough concentration was not within the therapeutic target, and 120 C_min_s were selected, while the remaining 144 C_min_s were excluded because of admeasurement after withdrawal or lack of the corresponding key values of clinical or laboratory data. The results of bivariate correlation analysis show that sex, the genotype of *CYP2C19*, daily dose, prothrombin time activity (PTA), international normalized ratio (INR), platelet, and Model for end-stage liver disease (MELD) score are significant factors. Further, daily dose and platelet are closely correlated with voriconazole concentration (absolute value of coefficient >0.3) (Table 3).

Artificial extracorporeal liver support, CYP2C19 inhibitors, and antimicrobial agents, including proton pump inhibitors, antibacterial and antifungal agents, did not affect subsequent voriconazole trough concentrations (*p* = 0.540, 0.962, and 0.775, respectively). Similarly, there were no significant associations between patient age or weight and voriconazole trough levels (Table 3).

Subsequently, the statistically significant factors entered into a stepwise multiple linear regression model of 115 C_min_s, specifically, daily dose, *CYP2C19* genotyping, sex, and PTA. Moreover, these factors were not collinear with one another (VIF < 5) considered independent and not confounding. Daily dose contributed the most to the voriconazole concentration, followed by PTA, sex, and genotype of *CYP2C19* (Table 4). The final results showed that the model fitted well (R^2^ = 0.348) and was statistically significant (*p* < 0.001).

On average, compared to the concentration of patients in *CYP2C19*1/*1*, the concentration tended to be 1.308 mg/L higher in *CYP2C19*2/*2* patients (*p* = 0.014), whereas no significant change in other genotyping (*p* = 0.535, 0.09). Moreover, the C_min_s would increase by 1.3 mg/L as the dosage increase by 100 mg. By contrast, male patients would decrease 1.651 mg/L compared to the female. Furthermore, PTA count was implicated with the voriconazole trough concentration, as well. The concentration would decrease by 0.035 mg/L with a one-unit increase in PTA count (Table 4).

The multiple linear regression equation was as follows:C_min_ = 0.012* daily does + *CYP2C19** (0.266* A *+* 1.252* B *+* 1.492* C) + 1.602*gender* D-0.036* PTA + 1.760(1)

(“A = 1, B = 1or C = 1” if the patient is identified as *CYP2C19 *1/*2, *1/*3, *2/*2*, respectively, otherwise “A = 0, B = 0 or C = 0”; “D = 1” if the patient is male, otherwise “D = 0”).

(Compared to *CYP2C19*1/*1*, dealt with the operation of dummy variables).

## 3. Discussion

Patients with CP-C cirrhosis are susceptible to fungal infection, a major cause of morbidity and mortality in cirrhosis [25]. Voriconazole is a broad-spectrum triazole antifungal agent with good in-body activity against *Aspergillosis* bacteria, fluconazole-resistant *Candida* [11]. It is commonly used clinically to prevent and treat IFIs, recommended as a first-line drug for invasive aspergillosis by IDSA and CPIC [11,12].

The oral bioavailability of voriconazole was 91.6%, approaching that of intravenously, and the intravenously and orally showed similar efficacy and safety in patients with severe hepatic dysfunction [26], whereas the quality of evidence was inferior and not for cirrhosis patients [27]. Although we observed 23.3% of patients received transform from intravenously to orally, the present study did not study the effect of the administration route for CP-C patients.

Targeted concentrations were recommended between 1.0–5.5 mg/L by British guidelines and other studies [13,14]. In this study, the C_1_ was higher than C_0_ and C_ss_ generally. Only 9.3% (4/43) patients always got the targeted trough concentrations after antifungal therapy, and 62.8% of patients (27/43) applied to adjustment. It is noteworthy that among the 31 (72.1%) patients with targeted C_0_, the subsequent concentrations of 25.8% (8/31) patients went above the target (>5.5 mg/L) on the unchanged regimen, implying accumulation. It was consistent with the results on voriconazole median terminal t_1/2_ and clearance of CP-C patients in ICU (60.7 h and 2.04 L/h, respectively) [24]. These results mean that repeated TDM is necessary. While among the 12 patients, 66.6% (8/12 patients) achieved the therapeutic target after adjusting. The majority had timely withdrawal or the targeted final C_min_s, whether for 12 patients with unideal C_0_, 8 patients with unideal C_1_, or 37 CP-C patients measured C_ss_. Above all, we conclude that voriconazole is prone to vary even in the same patients with CP-C. Voriconazole displays high inter-individual variability, and the administration in therapeutic doses leads to extremely varied serum levels from patient to patient and even in the same patient [28]. Voriconazole concentrations were closely related to efficacy and safety. Therefore, TDM is essential to provide patient-specific dosing recommendations, leading to more effective antifungal regimens to increase clinical efficacy and reduce adverse drug reactions [29]. We recommend more cautious administration for CP-C patients and more timely and appropriate adjustments in the clinic.

Voriconazole is metabolized primarily by CYP2C19, and the previous research reported on the low frequency of the *CYP2C19*17* allele, an ultrarapid metabolizer genotype, in Chinese subjects (<5%) relative to Ethiopians and Swedes (18% in both) [30,31]. Similarly, this study only collected one patient with *CYP2C19 *1/*17*, and for the sake of simplicity, we considered him within the same group as the wild-type *CYP2C19*1* homozygote, similar to the classification of Sugimoto et al. [32]. Simultaneously, the *CYP2C19*17* allele’s clinical relevance should not yet be discounted because the available evidence suggests that it predicted both voriconazole exposure and the dose required to achieve effective and non-toxic concentrations [33,34]. The present study investigated the significant factors, sex, genotype of *CYP2C19*, daily dose, PTA, INR, platelet, and MELD score associated with the variability of voriconazole trough concentrations (*n* = 117). Interestingly, the MELD score is another score of end-stage liver disease and is calculated by PTA value, related to INR, affecting our result. Shi C drew the same conclusion of the correlation between the genotype of *CYP2C19* and concentration [20]. The other research team members have drawn a similar conclusion for platelet count: platelet count was statistically significantly associated with voriconazole pharmacokinetics [35]. Except for the above factors, the variability of concentrations can be partly explained by non-linear kinetics, weight, liver function, concomitant medications [20], clinical and laboratory data such as C-reactive protein [17,36], albumin [37], and hemoglobin [38]. The different findings from these studies and the influence of sex found by this one may be due to the limited sample size and the large proportion of male patients (90.7%). Although we did not find the effect of artificial extracorporeal liver support and CYP2C19 inhibitors in this specific population, further research is needed, considering that CYP2C19 inhibitors can influence voriconazole exposure to a great extent theoretically [39] and the effect on drug metabolism and excretion of artificial extracorporeal liver support is still unclear [40].

Furthermore, the multiple linear regression model (1) is meaningful (*p* < 0.001). Daily dose contributed the most to the voriconazole concentration, followed by PTA, sex. and genotype of *CYP2C19* according to the standardized coefficients, without collinear with each other (VIF < 5). *CYP2C19*2/*2*, the poor metabolizer, is related to the higher concentration, implying that *CYP2C19* testing is still meaningful for precision medicine of Child–Pugh class C cirrhosis patients. Though there is insufficient evidence on the relationship between *CYP2C19* genotypes and clinical outcomes, there is a great potential for the initial voriconazole dose selection to be guided by the *CYP2C19* genotype [41]. Furthermore, more and larger studies, including multiple doses, are needed to confirm the impact of pharmacogenetics on voriconazole. The equation fits well and explains the 34.8% variety of voriconazole trough concentrations (R^2^ = 0.348), which means remaining relatively high unexplained variability and demanding further improvement. What is more, the pharmacokinetic characteristics of voriconazole can be impaired in patients with decompensated cirrhosis, which is related to factors such as affecting plasma protein binding rate or reducing the blood/plasma clearance [42], resulting in an increased risk of voriconazole accumulation and subsequent adverse events, but the separate influence of clinicopathological variables on the expression of hepatic CYP450 proteins has not been well characterized [43]. It is worthy of attention to the relationship between liver cirrhosis and the expression of CYP2C19.

The results showed significant differences in the three groups’ corresponding concentration by daily dose (*p* = 0.003). Administration dosage is a major essential factor contributing to the highly variable concentrations of voriconazole. However, the medication label of voriconazole does not recommend for individuals with CP-C cirrhosis, which means the empirical regimen is the common choice in the clinic, and it is urgent and vital to find a rational administration in such patients. The present study results imply that the administration regimen may be inappropriate in patients with CP-C cirrhosis, concretely, a loading dose regimen of 400 mg and maintenance dose of 100 mg twice daily or 200 mg daily orally or intravenously because of the higher voriconazole trough concentration [44]. We recommend the half-loading dose and far less maintenance dose than that of the patients with Child–Pugh class A or B, and 100 mg or less daily orally or intravenously is a better choice. Similar to voriconazole, Isavuconazole, a novel triazole antifungal agent used as a fourth-line agent [45], is also exclusively metabolized by the liver and characterized by a high half-life in patients with severe hepatic impairment. However, unlike voriconazole, the PPK model of Isavuconazole included two compartments [46], and no dosage adjustment for Isavuconazole is required for mild to moderate hepatic impairment [47], which is with fewer adverse events and fewer drug interactions but more frequent late-onset liver toxicity than voriconazole [48]. Dialectically, considering the wide range of C_min_s in each genotypic group and overlapping concentrations (Figure 1 and Figure 2a) and poorer live function of CP-C patients than others, we doubt that dosing based on *CYP2C19* polymorphism. It is suggested that the adjustment of the administration regimen should be mainly based on the results of TDM, while the genotype is only a reference. There are several limitations to the present study, although it is prospective. Firstly, this study has a small sample size, and it is a single-center study. Secondly, because only one patient had the genotype of CYP2C19*1*17, we could not conduct any statistical analysis for this individual but considered him within the group of the wild-type CYP2C19*1 homozygote. Thirdly, there is a lack of comparison with how other antifungals act. Thus, further well-designed prospective studies are necessary, and we also need to conduct system assessment about pharmacokinetic-pharmacodynamic analysis to gain more and better evidence to recommend adequate dosing regimens for adult patients with severe hepatic cirrhosis. Clinically, meticulous monitoring voriconazole trough concentration constantly during therapy is highly needed in patients with Child–Pugh class C cirrhosis to increase clinical efficacy and avoid adverse reactions.

## 4. Materials and Methods

### 4.1. Patients

Researchers conducted the prospective observational study and collected the hospital information and laboratory information at one tertiary hospital from January 2018 to December 2019. Patients with CP-C treated by voriconazole for either prophylaxis or treatment of a suspected fungal infection were identified and then screened by applying inclusion criteria of (1) diagnosed with liver dysfunction, such as liver failure or liver cirrhosis according to the Child–Pugh classifications, (2) performed therapeutic drug monitoring (TDM) during voriconazole therapy at least three days after initiation of a loading dose or maintenance or an adjusted dose of four days and retested about every three days [26], (3) aged 18 years and older, and exclusion criteria of (1) pregnancy or lactation and (2) missing the key values of voriconazole-related information, clinical data, or laboratory data. The administration regimen is mainly under empirical treatment regimens, and the adjustment is generally based on the results of TDM. This study was approved by the Ethics Committee of The Second XiangYa Hospital of Central South University with an approved NO. yxlb-lays-2018012.

### 4.2. Sample Collection, Storage and Bio-Analysis

Venous blood samples (2 mL) were collected into anticoagulant tubes. In addition, the steady-state of C_min_ of VRZ is recommended to be monitored, so samples were collected at least 3 days after initiation of a loading dose or maintenance and an adjusted dose of 4 days as well as within 2 h before any maintenance doses [26]. Monitoring continued every 3 days. All voriconazole plasma concentrations were analyzed by automatic two-dimensional liquid chromatography (2D-HPLC, Demeter Instrument Co., Ltd., Hunan, China) [35]. The two-dimensional separation conditions consisted of FRO C18 (100 mm × 3.0 mm, 5 μm, ANAX) and ASTON HD C18 (150 mm × 4.6 mm, 5 μm, ANAX) with the same flow rate 1.0 mL/min. The intra-day and inter-day precisions were 1.94% to 2.22% and 2.15% to 6.78%, respectively. The absolute and relative recovery ranged from 88.2% to 93.6% and 94.2% to 105.3%. This method is commonly applied in the clinical and passes the external quality assessment (EQA) annually. The stability of the blood sample at room temperature for 8 h and at −20 °C of 3 repeated freeze-thaw cycles was within ±8% and ±10%, respectively. This laboratory participated in the annual national external quality assessment scheme.

### 4.3. Genotype of CYP2C19

Genomic DNAs Deoxyribonucleic acid (DNA) of *CYP2C19* was extracted from whole blood samples (1–3 mL) using commercially available EZNA^®^ SQ. Blood DNA Kit II. Subsequently, the Sanger dideoxy DNA sequencing method was used for *CYP2C19* genotyping using the ABI3730xl fully automatic sequencing instrument (ABI Co.) from Boshang Biotechnology Co. Ltd. (Shanghai, China).

### 4.4. Data Collection

Data were extracted from medical records and laboratory information systems, including voriconazole-related information (C_min_s, timing, amount, route), demographics, laboratory data, and concomitant medications (antimicrobial agents, CYP2C19 inducers, and inhibitors). The first trough concentration, the repeated measured ones and the last one of patients were recorded as C_0_, C_1_, C_ss_, respectively. The Child–Pugh class and MELD score were calculated based on previous criteria.

### 4.5. Statistical Analysis

Data were presented as the mean ± standard (range) for continuous variables. The box plot showed by median and quartile.

The normality of quantitative data was tested using the Shapiro–Wilk test or Kolmogorov–Smirnov according to the sample size. On the basis of the result of normality, the statistical approach chosen was one of the four, the Student’s *t*-test, the nonparametric Kruskal–Wallis test, Mann–Whitney U test, or one-way ANOVA, to compare voriconazole trough concentrations, as appropriate.

The Pearson correlation and nonparametric Spearman correlation were used to study the relationship between clinical or laboratory data and C_min_s during the therapy, according to the type of variables. Then we identified the factors associated with variability of the voriconazole trough level. Statistically significant and independent factors correlated to voriconazole trough concentrations were filtered out to participate in the subsequent multiple linear regression and identified contributing to the variability in C_min_s. The C_min_s measured after withdrawal or lack of the corresponding key values of clinical or laboratory data were excluded in pairs in the bivariate correlation analysis or in rows in the multiple linear regression. For the regression analysis, the genotype of *CYP2C19* was set as a dummy variable. A variance inflation factor (VIF) of >5 was considered indicative of multicollinearity.

Data analysis was performed using IBM SPSS Statistics version 25 (IBM, New York, NY, USA), and the figures were drawn using GraphPad Prism version 8 (San Diego, CA, USA). All tests were two-sided. *p* values of <0.05 were considered significant.

## 5. Conclusions

This study explored predictors related to voriconazole administration regimen clinically and the change of trough concentrations in different periods and their effects for individuals with Child–Pugh class C cirrhosis in the prospective fashion.
Sex, *CYP2C19* genotyping, daily dose, PTA, INR, platelet, and MELD score correlated with the measured trough concentrations.Daily dose, PTA, sex and genotype of *CYP2C19* entered into a stepwise multiple linear regression about concentrations, and daily dose contributed the most. *CYP2C19*2/*2* contributed high concentrations, meaning that *CYP2C19* testing for precision medicine of Child–Pugh class C cirrhosis patients.Great importance was attached to therapeutic drug monitoring constantly during voriconazole therapy. In conclusion, it needs more cautious administration clinically due to no recommendation for Child–Pugh class C patients in the medication label. The adjustment of the administration regimen should be mainly based on the results of repeated therapeutic drug monitoring.

## Figures and Tables

**Figure 1 antibiotics-10-01130-f001:**
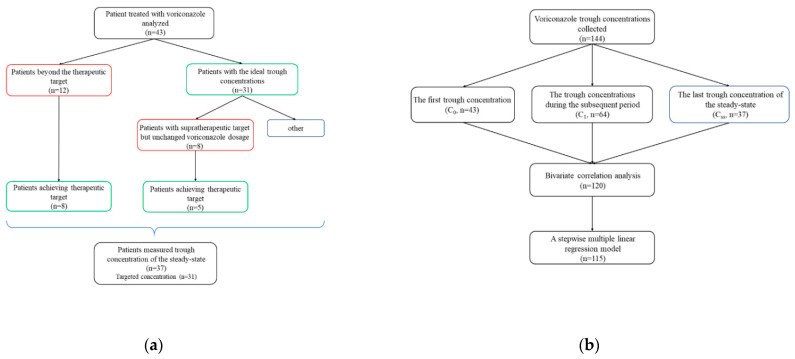
The flow charts show (**a**) changes of voriconazole trough concentrations in 43 patients during treatment clinically; (**b**) statistical analysis of 144 measured voriconazole trough concentrations.

**Figure 2 antibiotics-10-01130-f002:**
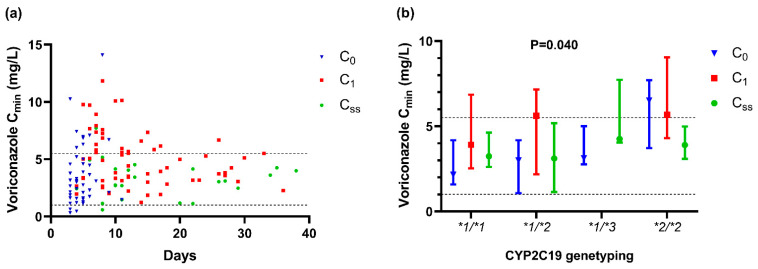
Distinction of voriconazole trough concentration of 43 patients (**a**) as the therapeutic progress by days (*n* = 144); (**b**) in different *CYP2C19* phenotype groups (*n* = 103, *p* = 0.040). The Kruskal–Wallis test was used to compare C_min_s of the four groups. Data are expressed as the median ± interquartile range. C_min_, voriconazole trough concentration; *CYP2C19*, cytochrome P450 2C19; C_0_, the first trough concentration after voriconazole therapy; C_ss_, the final trough concentration; C_1_, the repeated measured trough concentration except C_ss_. **1*, **2*, **3*, **17* reprented the single nucleotide sequence of genotype of *CYP2C19*.

**Figure 3 antibiotics-10-01130-f003:**
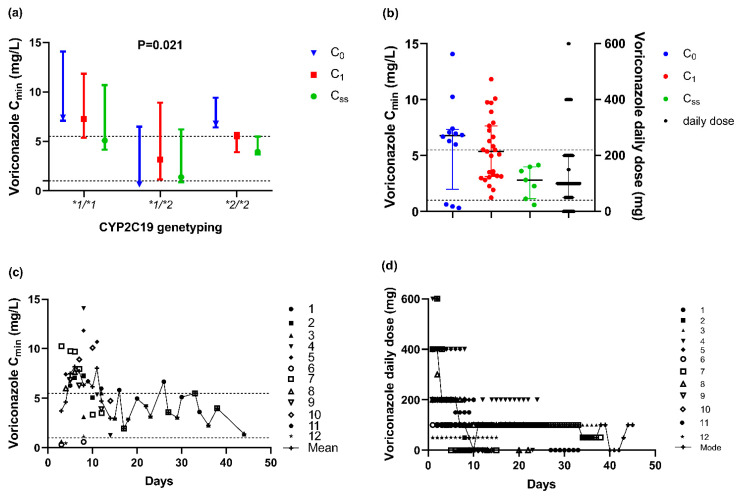
For 12 patients started beyond the therapeutic target (**a**) Distinction of voriconazole trough concentration in different *CYP2C19* phenotype groups (*n* = 50, *p* = 0.021). The Kruskal–Wallis test was used to compare C_min_s. (**b**) Distinction of voriconazole trough concentration in different groups of C_0_, C_1_, C_ss_ (*n* = 50), and daily dose (*n* = 27). Data are expressed as the median ± interquartile range. (**c**) The mean concentration of such 12 patients changed by days. (**d**) Every dosage adjustment (*n* = 247) and the mode dose (*n* = 45) by days (It showed the highest one if there were more than one mode). **1*, **2*, **3*, **17* reprented the single nucleotide sequence of genotype of *CYP2C19*.

**Figure 4 antibiotics-10-01130-f004:**
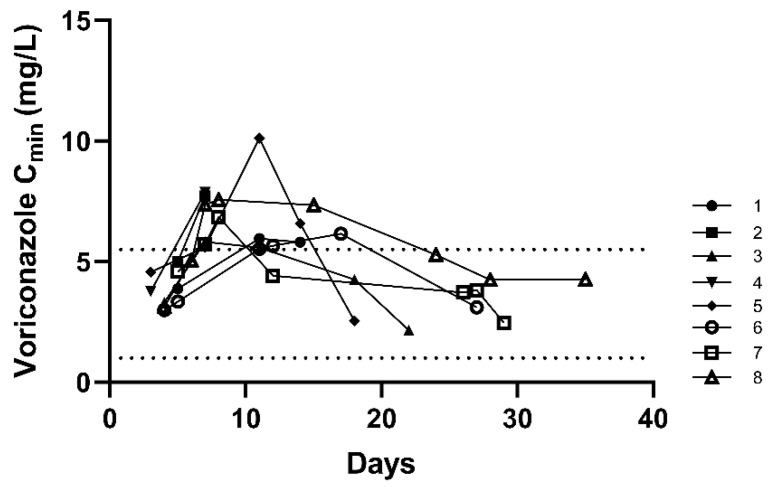
The voriconazole trough concentrations by days of 8 patients who had targeted C_0_ but supratherapeutic C_1_ during the constant unchanged therapy dosage of voriconazole (*n* = 38).

**Figure 5 antibiotics-10-01130-f005:**
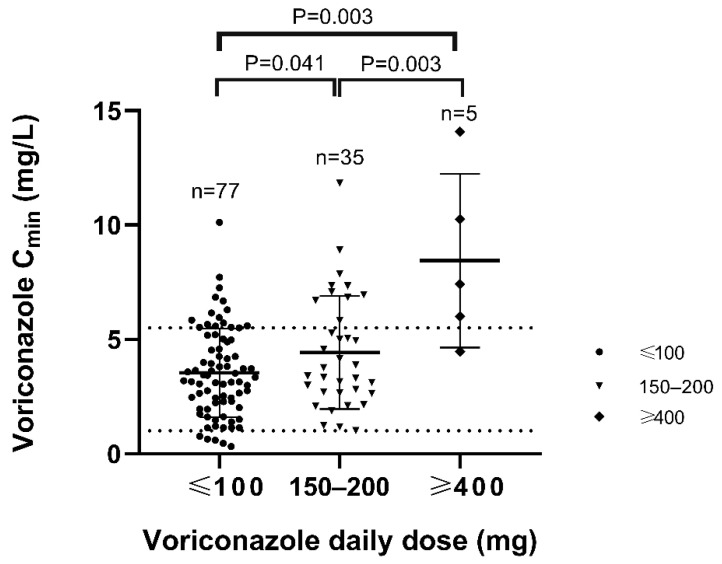
Distribution of voriconazole trough levels over daily dosages. The numbers of measurements for each daily dose are reported.

**Table 1 antibiotics-10-01130-t001:** Demographic, clinical data and voriconazole regimen for 43 CP-C patients with therapeutic drug monitoring of voriconazole.

Variable	Value
Mean ± S (range) age (years)	49.35 ± 11.65 (32–89)
Sex (no. [%] of males)	39 (90.7)
Mean ± S (range) weight (kg)	61.27 ± 12.87 (36–99)
No. (%) Genotype of *CYP2C19*	
**1/*1*	20 (46.5)
**1/*17*	1 (2.3)
**1/*2*	13 (30.2)
**1/*3*	3 (7.0)
**2/*2*	6 (14.0)
No. (%) route of administration before C_0_	
intravenously	15 (34.9)
orally	23 (53.5)
in sequential therapy	5 (11.6)
No. (%) transform from intravenously to orally throughout the treatment	10 (23.3)
Median sampling time of C_0_ (IQR, range)	5 (3–5, 3–11days)
Concentrate (mg/mL)	
Median (IQR, range) voriconazole trough level (mg/L)	3.745 (2.485–5.6425, 0.32–14.08) *n* = 144
C_0_	2.99 (1.61–5.00, 0.32–14.08) *n* = 43
C_1_	4.33 (3.0775–6.1000, 1.86–11.83) *n* = 64
C_ss_	3.90 (2.51–4.84, 0.60–10.70) *n* = 37
Proportion (%) of targeted C_min_s (mg/L) ^1^	
C_0_	31/43 (72.1)
C_ss_ ^2^	31/37 (83.8)
C_ss1_ ^3^	8/12 (66.6)
C_ss2_ ^4^	5/8 (62.5)
No. (%) of adjustment times	
0	16 (37.2)
1	11 (25.6)
2	7 (16.3)
3–7	9 (21.9)
No. (%) of concomitant medication ^5^	37 (86.0)
CYP2C19 inhibitors	31 (72.1)
Antimicrobial	25 (58.1)
CYP3A4 inhibitors	1 (2.3)

^1^ Targeted concentrations were recommended between 1.0–5.5 mg/L. ^2^ C_ss_ was available from 37 patients with CP-C cirrhosis, while the remaining 6 patients had no C_ss_. ^3^ C_ss1_ was evaluated in 12 patients with untargeted C_0_. ^4^ C_ss2_ was evaluated in 8 patients with supratherapeutic concentration, whereas no adjusted daily dose was applied in their subsequent therapy. ^5^ Concomitant medication was applied to 37 out of 43 CP-C patients. CYP, cytochrome P450; C_min_s, voriconazole trough concentration; C_0_, the first trough concentration after voriconazole therapy; C_ss_, the final trough concentration; C1, the repeated measured trough concentration except C_ss_; IQR, interquartile range. **1*, **2*, **3*, **17* reprented the single nucleotide sequence of genotype of *CYP2C19*.

**Table 2 antibiotics-10-01130-t002:** Administration dosage of voriconazole in 43 CP-C patients ^2^.

Regimen	Value
No. (%) of Loading dosage (mg) ^1^	22 (51.2)
400/12 h	5 (22.7)
200/12 h	4 (18.2)
200/24 h	4 (18.2)
No. (%) of Maintain dosage (mg)	
200/12 h 200/24 h 100/24 h 100/12 h	8 (18.6) 8 (18.6) 10 (23.3) 9 (20.9)
No. (%) of final steady-state administration (mg)	
200/24 h 100/24 h	8 (18.6) 15 (34.9)
50/12 h	8 (18.6)

^1^ Loading dosage was applied to 22/43 patients. ^2^ The 3 most widely applied regimens in 43 patients with CP-C cirrhosis are shown.

**Table 3 antibiotics-10-01130-t003:** Multivariate bivariate correlation analysis of factors associated with voriconazole plasma concentration.

Variable	Coefficient	*p*-Value
Age	−0.064	0.485
Sex	0.221	0.015 *
Weight	0.001	0.993
Daily dose	0.329	<0.001 *
*CYP2C19* genotyping ^1^		
**1/*2*	−0.068	0.417
**1/*3*	0.196	0.018 *
**2/*2*	0.216	0.009 *
Platelet	−0.302	0.001 *
INR	0.184	0.047 *
PTA	−0.278	0.002 *
MELD score	0.184	0.048 *
Hemoglobin	−0.063	0.518
Alanine aminotransferase	−0.037	0.696
Aspartate aminotransferase	0.051	0.588
Total bilirubin	0.104	0.265
Direct bilirubin	0.092	0.324
Bile acid	0.016	0.862
Albumin	0.153	0.101
Blood urea nitrogen	0.058	0.532
Creatinine	0.044	0.638
Creatinine Clearance	0.010	0.915
Prothrombin time	0.153	0.100
Artificial extracorporeal liver	0.044	0.635
Concomitant agents ^2^		
CY2C19 inhibitors	0.048	0.609
Antimicrobial agents	−0.010	0.919

^1^*CYP2C19* genotyping was dealt with the operation of dummy variables compared to *CYP2C19*1/*1*. We considered the patient with *CYP2C19 *1/*17* (*n* = 1) within the same group as the *CYP2C19*1/**1. ^2^ CYP2C19 inducers were not analyzed as only one patient applicated. IQR, interquartile range; PTA, prothrombin time activity; INR, international normalized ratio; MELD, Model for end-stage liver disease. *p* < 0.05 and variables with statistical significance are expressed as *.

**Table 4 antibiotics-10-01130-t004:** The multiple linear regression model about voriconazole trough concentration ^2^.

Variable	Unstandardized Coefficients		Standardized Coefficients	t	*p*-Value	VIF
	B	Std. Error				
Intercept	1.760	0.849	-	2.072	0.041	-
Daily dose	0.012	0.002	0.439	5.236	0.000	1.069
PTA	−0.036	0.012	−0.238	−3.139	0.002	1.038
Sex	1.602	0.598	−0.217	2.645	0.009	1.064
*CYP2C19* genotyping ^1^						
**1/*2*	0.266	0.428	0.053	0.622	0.535	1.192
**1/*3*	1.252	0.733	0.140	1.708	0.090	1.121
**2/*2*	1.492	0.596	0.209	2.503	0.014	1.166
F = 9.686				<0.001		

^1^ Compared to *CYP2C19*1/*1*, dealt with the operation of dummy variables. ^2^ R^2^ = 0.348; Durbin–Watson test value = 1.801; 115 voriconazole trough concentrations were selected and analyzed.

## Data Availability

I declare that my research data is available. I will share my research data with other researchers if they need it. Especially, the data will include (but is not limited to): raw data, processed data, software, algorithms. If additional files are required, they will also be shared on request. Meanwhile, the data will become available since the date it was published. The researchers can email me if they are interested in the study and need the research data for analysis.

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
