# Peer review of "Predictors of Voriconazole Trough Concentrations in Patients with Child–Pugh Class C Cirrhosis: A Prospective Study"

_antibiotics, 2021, doi:10.3390/antibiotics10091130_

Round 1

Reviewer 1 Report

Dear Authors, 

your manuscript is well written and detailed. Please, recheck it , because it contains many typo's (Child-Pugh/Child-pugh etc). 

In the discussion section, I do not see as clearly stated what the limitations of this study are: low numbers, lack of comparison with other antifungals act.

In particular, what about any differences with the new isavuconazole in patients with liver injury? (Pagano et al, Journal of Fungi, Desai et al, AAC).

Reviewer 2 Report

In my opinion the manuscript entitled „Predictors of Voriconazole Trough Concentrations in Patients  with Child-Pugh Class C Cirrhosis: A Prospective Study” presents an interesting topic in the field of cirrhosis therapy. The aim of this study was to investigate the variability of trough concentration of voriconazole administrations in patients with cirrhosis. In my opinion, the presented manuscript requires minor corrections before publication in the journal. The manuscript has been prepared very carefully. The Introduction section is a very good justification and explanation of the research undertaken by Authors. I have one comment - please complete this section with statistical data on the frequency of the Child-Pugh Class C Cirrhosis. How important is this public health problem? Especially that the characteristics of the patients and the method of their recruitment are in two different places in the manuscript, which makes it difficult to understand the procedure. The results were clearly presented and their description does not raise any objections. Minor editorial note: In table 4, the title should be above not below the table. Authors discuss the obtained results in the Discussion section. In my opinion this part of the manuscript is very good. However, I have doubts about the detailed results of statistical analyzes given by Authors in this section. I guess they shouldn't be in the discussion. I think Authors should also present the limitations of their study. Moreover, Authors may provide also information about the possible future research directions. However, I rate the manuscript very highly and  congratulations to Authors of their well-prepared text.

Reviewer 3 Report

Zhao et. al., have presented a prospective study describing voriconazole trough concentrations in patients with child-pugh class c cirrhosis. As voriconazole is a narrow therapeutic index drug and has shown large intra- and inter-individual PK variability, often requires TDM in patients. This would be useful information for physicians treating the child-pugh class C patients with voriconazole. The study is straightforward. The article reads well and is generally well written, there are occasional lapses in grammar/typos which should be corrected by a thorough re-reading and revision of the manuscript. However, there are a few points that could be considered:

  1. It is suggested to include more information on the voriconazole pharmacokinetics, especially the route of elimination, the fraction of the drug eliminated unchanged through renal excretion, the fraction metabolized of voriconazole by CYP2C19, etc

  1. Does liver cirrhosis impact the expression of CYP2C19? If this information is available please include it.

  1. Were there any special considerations for selecting the loading and maintenance dose for these patients? Please comment.

  1. In contrast to the current finding, in general, it appears to be a poor correlation between the CYP2C19 genotyping and the PK variability of voriconazole. Please comment.

Zonios D, et. al. Voriconazole metabolism, toxicity, and the effect of cytochrome P450 2C19 genotype. J Infect Dis. 2014 Jun 15;209(12):1941-8. PMID: 24403552;

  1. Line 78: Correct “CPY2C19 inhibitors” to “CYP2C19 inhibitors”; Line 300: Correct “Pantients” to patient

  1. Suggest improving the quality of Figures 3c and 3d.
